# pFedSAM: Secure Federated Learning Against Backdoor Attacks via Personalized Sharpness-Aware Minimization

## Abstract

Federated learning is a distributed learning paradigm that allows clients to perform collaboratively model training without sharing their local data. Despite its benefit, federated learning is vulnerable to backdoor attacks where malicious clients inject backdoors into the global model aggregation process so that the resulting model will misclassify the samples with backdoor triggers while performing normally on the benign samples. Existing defenses against backdoor attacks either are effective only under very specific attack models or severely deteriorate the model performance on benign samples. To address these deficiencies, this paper proposes pFedSAM, a new federated learning method based on partial model personalization and sharpness-aware training. Theoretically, we analyze the convergence properties of pFedSAM for the general non-convex and heterogeneous data setting. Empirically, we conduct extensive experiments on a suite of federated datasets and show the superiority of pFedSAM over state-of-the-art robust baselines in terms of both robustness and accuracy.

## 1 Introduction

Federated learning (FL) has emerged as a transformative paradigm in machine learning, enabling collaborative model training among distributed clients while keeping their data locally. Although FL has achieved success in many applications, such as keyboard prediction Hard et al. (2018), medical image analysis Li et al. (2020), and Internet of things Samarakoon et al. (2018), FL systems confront significant security threats due to their distributed nature, particularly from backdoor attacks Bagdasaryan et al. (2020); Wang et al. (2020); Xie et al. (2020); Sun et al. (2019). Specifically, by stealthily injecting backdoor triggers into the trained model, attackers aim to mislead any input with the backdoor trigger to a target label while ensuring that the backdoored model's performance on benign samples remains unaffected. Such stealthy manipulation makes backdoor attacks one of the most serious threat to the real-world deployment of FL system.

Existing defenses against backdoor attacks in FL can be roughly divided into two categories Nguyen et al. (2022): anomaly update detection and robust federated training. The first category consists of anomaly detection approaches that can identify whether the submitted updates are malicious and then remove the malicious ones, such as Krum Blanchard et al. (2017), Trimmed Mean Yin et al. (2018), Bulyan Guerraoui et al. (2018), and FoolsGold Fung et al. (2018). However, these methods are effective only under very specific attack models (i.e., attack strategies of the adversary and data distribution of the benign clients). The second category comprises robust federated training methods that can directly mitigate backdoor attacks during the training process, such as norm clipping Sun et al. (2019); Xie et al. (2021) and adding noise Xie et al. (2021). These solutions require modification of the individual weights of benign model updates and therefore result in severe degradation on model performance on benign samples. Moreover, most of the aforementioned works only work in the single-shot attack setting where a small number of malicious clients participate in a few rounds but fail under the stronger continuous attack setting where malicious clients continuously participate in the entire FL training period Zhang et al. (2023).

A few recent works Qin et al. (2023); Li et al. (2021a); Lin et al. (2022) have demonstrated that personalized federated learning (pFL) methods that were originally designed to improve accuracy

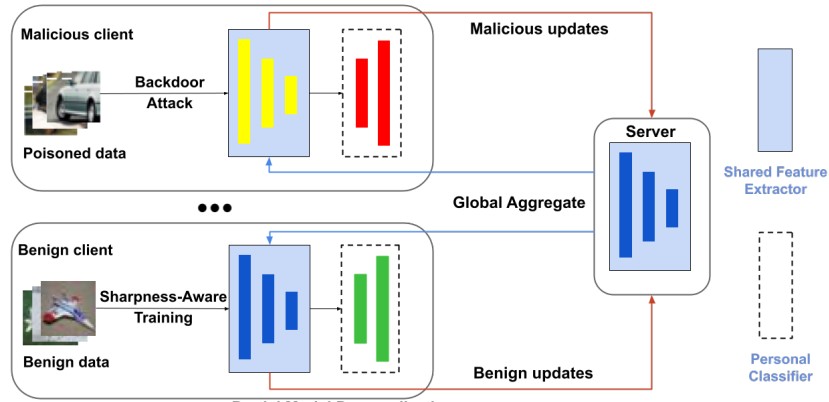

Figure 1: An overview of pFedSAM under backdoor attacks. Partial model personalization allows each client to locally retain a personal classifier and only share the feature extractor with the server for aggregation. The malicious client performs the backdoor attack and sends the malicious updates, while the benign client engages in sharpness-aware training and sends the benign updates to the server. The server aggregates the shared feature extractor and sends it back to all clients.

under heterogeneous data distribution could also provide some robustness benefits. Specifically, Li et al. (2021a) and Lin et al. (2022) utilize model personalization to defend against *untargeted* poisoning attacks that aim to corrupt FL models' prediction performance or make FL training diverge, but do not address the more challenging problem of backdoor attacks. Qin et al. (2023) further demonstrates that pFL with partial model-sharing can notably enhance robustness against backdoor attacks in comparison to pFL with full model-sharing under the continous attack setting, but it solely focuses on the black-box setting where malicious adversaries can only manipulate training data and have no control of the training process. Considering the white-box setting where malicious clients can control the local training process, Ye et al. (2023) demonstrates pFL methods with partial model-sharing remain vulnerable to backdoor attacks. Therefore, a straightforward implementation of pFL is susceptible to new attacks tailored for pFL and does not ensure robustness against real-world backdoor attacks.

In this paper, we propose pFedSAM, a novel personalized FL method that can inherently defend against both black-box and white-box state-of-the-art backdoor attacks while maintaining the benign performance of the models. This is achieved by two key modules: partial model personalization and sharpness-aware training. The partial model personalization lets each client own its locally preserved linear classifier to block the propagation of backdoor features from malicious clients to benign clients. The sharpness-aware training generates local flat model updates with better stability and perturbation resilience, resulting in a globally flat model that is robust to the injection of backdoor features from malicious clients. The overview of pFedSAM is shown in Figure 1. We summarize our main contributions as follows.

- We propose pFedSAM, a novel pFL method that offers better robustness against both black-box and white-box backdoor attacks while retaining similar or superior accuracy on the benign model performance relative to other common robust FL methods.

- We provide convergence guarantees for our proposed pFedSAM method under the general non-convex and non-IID data distribution setting.

- We conduct an extensive evaluation of the proposed method on several FL benchmark datasets by comparing it with state-of-the-art baselines under the stronger and stealthier continuous black-box and white-box backdoor attacks. The empirical results show that the proposed method can largely outperform the baselines in terms of both attack success rate and main task accuracy.

## 2 BACKGROUND AND RELATED WORKS

### 2.1 PERSONALIZED FEDERATED LEARNING

We consider a typical FL system involving $N$ clients and one server. Each client $i \in [N]$ holds a local training dataset $\mathcal{D}_i = \{\xi_{i,j}\}_{j=1}^{D_i}$ where $\xi_{i,j}$ is a training example and $D_i$ is the size of the local training dataset. The total number of training examples across $N$ devices is $D = \sum_{i=1}^{N} D_i$. Let $w \in \mathbb{R}^d$ denote the parameters of a machine learning model and $f_i(w, \xi_{i,j})$ be the loss of the model on the sample $\xi_{i,j}$. Then the loss function of client $i$ is $F_i(w) = (1/D_i) \sum_{j \in \mathcal{D}_j} f_i(w_i, \xi_{i,j})$. The objective of standard FL is to find model parameters that minimize the weighted average loss over all clients:

$$\min_{\{w_i\}_{i=1}^{N}} \sum_{i=1}^{N} \alpha_i F_i(w), \tag{1}$$

where $\alpha_i > 0$ is the weight assigned to client $i$, and $\sum_{i=1}^{N} \alpha_i = 1$. However, standard FL can be ineffective and undesirable under data heterogeneity Li et al. (2021a). Instead, pFL aims to train local personalized models instead of a single global model across all clients, which is more adaptive to each client's local dataset and has shown to improve model accuracy under practical non-IID scenarios. Based on the form of model sharing with the server, existing pFL methods can be divided into two categories: full model-sharing and partial model-sharing.

- Full model-sharing Fallah et al. (2020); Li et al. (2021a); Liang et al. (2020); Marfoq et al. (2021); T Dinh et al. (2020): The objective can be summarized as

$$\min_{w_0, \{w_i\}_{i=1}^{N}} \sum_{i=1}^{N} \alpha_i \left( F_i(w_i) + \lambda_i R(w_0, w_i) \right), \tag{2}$$

where $w_0$ is a reference model shared among all clients, $w_i$ means the local personalized model owned by client $i$, and $\lambda_i$ is the weight of the regularization term $R(w_0, w_i)$ for client $i$.

- Partial model-sharing Arivazhagan et al. (2019); Collins et al. (2021); Li et al. (2021b); Pillutla et al. (2022): The objective can be summarized as

$$\min_{\phi, \{h_i\}_{i=1}^{N}} \sum_{i=1}^{N} \alpha_i F_i(\phi, h_i), \tag{3}$$

where the full model parameters $w_i$ of each client $i$ are divided into two parts: *shared* parameters $\phi \in \mathbb{R}^{d_0}$ and *personal* parameters $h_i \in \mathbb{R}^{d_i}$, i.e. $w_i = (\phi, h_i)$.

As shown in Pillutla et al. (2022), partial model-sharing personalization can obtain most of the benefit of full model-sharing personalization with only a small fraction of personalized parameters. Our work builds on FedRep Collins et al. (2021), a pFL algorithm with partial model-sharing that focuses on learning shared representations and personal classifier heads between clients but does not consider robustness. In contrast, our work provides a novel robust FL framework. Moreover, a major different ingredient of our algorithm is the sharpness-aware training for shared representation learning, which finds backdoor-resilient global shared parameters in each FL round.

### 2.2 BACKDOOR ATTACKS IN FEDERATED LEARNING

In FL backdoor attacks, the adversary controls a group of malicious clients to manipulate their local models, which are then aggregated into the global model and affect its properties. In particular, the adversary wants the backdoored global model to mislead the prediction on inputs with the backdoor trigger to a target label while behaving normally on all benign samples. There are generally two categories for FL backdoor attacks: 1) *black-box setting*, where malicious clients tamper with a fraction of their training data, also known as data poisoning, to inject a backdoor into their local models during the training Xie et al. (2020); Goldblum et al. (2022); Lyu et al. (2022); and 2) *white-box setting*, where the adversary poisons the training data of the malicious clients and manipulates their training processes by modifying the resulting uploaded models, also known as model poisoning, to maximize attack impact while avoiding being detected. Examples of white-box backdoor attacks include *constrain-and-scale attack* Bagdasaryan et al. (2020), *projected gradient descent attack with model replacement* Wang et al. (2020), DBA Xie et al. (2020), and BapFL Ye et al. (2023).

# 3 PFEDSAM: FEDERATED LEARNING WITH PERSONALIZED SHARPNESS-AWARE MINIMIZATION

## 3.1 PFEDSAM ALGORITHM

---

**Algorithm 1** pFedSAM

---

**Input:** Initial states $\phi^0$, $\{h_i^0\}_{i=1}^N$, client sampling ratio $r$, number of local iterations $\tau_h$, $\tau_\phi$, number of communication rounds $T$, learning rates $\eta_h$, $\eta_\phi$, and neighborhood size $\rho$
    **Output:** Personalized models $(\phi^T, h_i^T), \forall i \in [N]$.

1: **for** $t = 0, 1, \ldots, T-1$ **do**
2:     Server randomly samples a set of $rN$ clients $\mathcal{S}^t$.
3:     Server broadcasts the current global version of the shared parameters $\phi^t$ to all clients in $\mathcal{S}^t$.
4:     **for** each client $i \in \mathcal{S}^t$ in parallel **do**
5:         Initialize $h_i^{t,0} = h_i^t$
6:         **for** $s = 0, \ldots, \tau_h - 1$ **do**
7:             Compute stochastic gradient $\widetilde{\nabla}_h F_i(\phi^t, h_i^{t,s})$
8:             $h_i^{t,s+1} = h_i^{t,s} - \eta_h \widetilde{\nabla}_h F_i(\phi^t, h_i^{t,s})$
9:         **end for**
10:        Update $h_i^{t+1} = h_i^{t,\tau_h}$ and initialize $\phi_i^{t,0} = \phi^t$
11:        **for** $s = 0, \ldots, \tau_\phi - 1$ **do**
12:            Update shared parameters using SAM according to (6) and (7)
13:        **end for**
14:        Update $\phi_i^{t+1} = \phi_i^{t,\tau_\phi}$
15:        Client sends $\phi_i^{t+1}$ back to server
16:     **end for**
17:     **for** each client $i \notin \mathcal{S}^t$ **do**
18:         $h_i^{t+1} = h_i^t$
19:     **end for**
20:     Server updates $\phi^{t+1} = \frac{1}{rN} \sum_{i \in \mathcal{S}^t} \phi_i^{t+1}$
21: **end for**

---

We present the pFedSAM algorithm to solve Problem (3) and describe its detailed procedures in Algorithm 1. The overall training process of pFedSAM consists of updating two parts of a client's model in an alternating manner: local *personal* parameters $h_i$ and global *shared* parameters $\phi$.

At the beginning of each $t$-th round, the server randomly samples a subset of the clients $\mathcal{S}^t$ to join the learning process (line 2) and broadcasts the current global version of the shared parameters $\phi^t$ to clients in $\mathcal{S}^t$ (line 3). Then, each selected client $i \in \mathcal{S}^t$ performs local training in two stages. First, it fixes the local version of the shared parameters $\phi_i$ to be the received global one $\phi^t$ and then performs $\tau_h$ iterations of SGD to update the personal parameters $h_i$ (lines 5–9). Second, it fixes the personal parameters $h_i$ to be the newly updated one $h_i^{t,\tau_h}$ obtained from the first stage (line 10) and then updates the shared parameters $\phi_i$. Here, instead of seeking out shared parameters that simply have low training loss by minimizing $F_i(\phi, h_i)$, we propose to find shared parameters whose entire neighborhoods have uniformly low training loss. This can be formulated as solving the following Sharpness-Aware Minimization (SAM) problem that jointly minimizes the loss function and smooths the loss landscape:

$$\min_\phi \max_{\|\epsilon\|_2 \le \rho} F_i(\phi + \epsilon, h_i^{t+1}), \tag{4}$$

where $\rho$ is a predefined constant controlling the radius of the perturbation and $\|\cdot\|_2$ is a $l_2$-norm. Intuitively, through optimizing the objective (4), the resulting local version of the shared parameters $\phi_i$ has a smoother local loss landscape and exhibits inherent robustness to perturbations. Then by aggregating all the local models with smoother local loss landscape at the server, the flatness of the aggregated global model is boosted as well, making it more resilient to the injection of backdoor features from malicious clients.

To solve the min-max problem (4), we adopt the efficient and effective approximation technique proposed in Foret et al. (2021). Specifically, via the use of the first-order Taylor expansion of $F_i$, the

solution of the inner maximization problem is

$$\epsilon^*(\phi) \approx \underset{\|\epsilon\|_2 \le \rho}{\arg\max} \left\{ F_i(\phi, h_i^{t+1}) + \epsilon^T \nabla_\phi F_i(\phi, h_i^{t+1}) \right\} = \rho \frac{\nabla_\phi F_i(\phi, h_i^{t+1})}{\left\| \nabla_\phi F_i(\phi, h_i^{t+1}) \right\|_2} \tag{5}$$

Substituting (5) back into (4) and taking the differentiation w.r.t. $\phi$, we can obtain the approximate SAM gradient as $\nabla_\phi F_i(\phi, h_i^{t+1})|_{\phi + \rho \nabla_\phi F_i(\phi, h_i^{t+1})/\nabla_\phi F_i(\phi, h_i^{t+1})}$. Therefore, at the $s$-th local iteration of round $t$, SAM first computes partial stochastic gradient $\widetilde{\nabla}_\phi F_i(\phi_i^{t,s}, h_i^{t+1})$ and calculates the perturbation $\epsilon(\phi_i^{t,s})$ as follows:

$$\epsilon(\phi_i^{t,s}) = \rho \frac{\widetilde{\nabla}_\phi F_i(\phi_i^{t,s}, h_i^{t+1})}{\left\| \widetilde{\nabla}_\phi F_i(\phi_i^{t,s}, h_i^{t+1}) \right\|_2}. \tag{6}$$

Then the perturbation is used to update the shared parameters as follows:

$$\phi_i^{t,s+1} = \phi_i^{t,s} - \eta_\phi \widetilde{\nabla}_\phi F_i(\phi_i^{t,s} + \epsilon(\phi_i^{t,s}), h_i^{t+1}), \tag{7}$$

where $\eta_\phi$ is the learning rate. The same procedure repeats for $\tau_\phi$ local iterations (lines 10–13).

After local training, each selected client $i$ only sends the updated local version of the shared parameters $\phi_i^{t+1}$ to the server, which aggregates them from all selected clients to compute the global version of the shared parameters $\phi^{t+1}$ for the next round (line 20). The updated personal parameters $h_i^{t+1}$ are kept locally at the client to serve as the initialization when the client is selected for another round.

### 3.2 Convergence Properties of pFedSAM

In this section, we give the convergence results of pFedSAM. To simplify presentation, we denote $H = (h_1, \ldots, h_N) \in \mathbb{R}^{d_1 + \cdots + d_N}$. We consider a general setting with $\alpha_i = 1/N$ without loss of generality. Then our objective becomes

$$\min_{\phi, H} F(\phi, H) = \frac{1}{N} \sum_{i=1}^{N} F_i(\phi, h_i). \tag{8}$$

Before stating our theoretical results, we make the following assumptions for the convergence analysis.

**Assumption 1** (Smoothness). *For each $i \in [N]$, the function $F_i$ is continuously differentiable. There exist constants $L_\phi$, $L_h$, $L_{\phi h}$, $L_{h\phi}$ such that for each $i \in [N]$:*

- *$\nabla_\phi F_i(\phi, h_i)$ is $L_\phi$-Lipschitz with respect to $\phi$ and $L_{\phi h}$-Lipschitz with respect to $h_i$, and*

- *$\nabla_h F_i(\phi, h_i)$ is $L_h$-Lipschitz with respect to $h_i$ and $L_{h\phi}$-Lipschitz with respect to $\phi$.*

*The relative cross-sensitivity of $\nabla_\phi F_i$ with respect to $h_i$ and $\nabla_h F_i$ with respect to $\phi$ is defined by the following scalar:*

$$\chi := \max\{L_{\phi h}, L_{h\phi}\}/\sqrt{L_\phi L_h}. \tag{9}$$

**Assumption 2** (Bounded Variance). *The stochastic gradients in Algorithm 1 are unbiased and have bouned variance. That is, for all $\phi$ and $h_i$,*

$$\mathbb{E}[\widetilde{\nabla}_\phi F_i(\phi, h_i)] = \nabla_\phi F_i(\phi, h_i), \tag{10}$$

$$\mathbb{E}[\widetilde{\nabla}_h F_i(\phi, h_i)] = \nabla_h F_i(\phi, h_i). \tag{11}$$

*Furthermore, there exist constants $\sigma_\phi$ and $\sigma_h$ such that*

$$\mathbb{E}\left\| \widetilde{\nabla}_\phi F_i(\phi, h_i) - \nabla_\phi F_i(\phi, h_i) \right\|^2 \le \sigma_\phi^2, \tag{12}$$

$$\mathbb{E}\left\| \widetilde{\nabla}_h F_i(\phi, h_i) - \nabla_h F_i(\phi, h_i) \right\|^2 \le \sigma_h^2. \tag{13}$$

Assumptions 1 and 2 are standard in the analysis of SGD Bottou et al. (2018), Guo et al. (2022). Here, we can view $\nabla_\phi F_i(\phi, h_i)$, when $i$ is randomly sampled from $[N]$, as a stochastic partial gradient of $F(\phi, H)$. The following assumption imposes a constant variance bound.

**Assumption 3** (Partial Gradient Diversity). *There exist a constant $\delta$ such that for all $\phi$ and $H$,*

$$\frac{1}{N}\sum_{i=1}^{N}\|\nabla_\phi F_i(\phi, h_i) - \nabla_\phi F_i(\phi, H)\|^2 \le \delta^2 \tag{14}$$

We denote $\Delta F_0 = F(\phi^0, H^0) - F^*$ with $F^*$ being the minimal value of $F(\cdot)$. Further, we use the shorthands $H^t = (h_1^t, \cdots, h_N^t)$, $\Delta_\phi^t = \|\nabla_\phi F(\phi^t, H^t)\|^2$, and $\Delta_h^t = 1/n \sum_{i=1}^{N} \|\nabla_h F(\phi^t, h_i^t)\|^2$.

Next, we propose our main theoretical results of the proposed pFedSAM algorithm in the following theorem.

**Theorem 1** (Convergence of Algorithm 1). *Under Assumptions 1-3, if the learning rates satisfy $\eta_\phi = \alpha/(L_\phi \tau_\phi)$ and $\eta_h = \alpha/(L_h \tau_h)$, where $\alpha$ depends on the parameters $L_\phi, L_h, \chi^2, \sigma_\phi^2, \sigma_h^2, r$, and the number of total rounds $T$, we have*

$$\frac{1}{T}\sum_{t=0}^{T-1}\left(\frac{1}{L_\phi}\mathbb{E}[\Delta_\phi^t] + \frac{r}{L_h}\mathbb{E}[\Delta_h^t]\right) \le \frac{(\Delta F_0 \Omega_1^2)^{1/2}}{\sqrt{T}} + \frac{(\Delta F_0^2 \Omega_2^2)^{1/3}}{T^{2/3}} + \mathcal{O}(\frac{1}{T}), \tag{15}$$

*where the effective variance terms are defined as follows:*

$$\Omega_1^2 = \frac{\sigma_h^2}{L_h}(r + \chi^2(1-r)) + \frac{\sigma_\phi^2}{L_\phi} + \frac{\delta_\phi^2}{L_\phi}(1-r),$$

$$\Omega_2^2 = \frac{\chi^2 \sigma_h^2}{L_h} + \frac{\rho^2}{\tau_\phi} + \frac{\sigma_\phi^2 + \delta^2}{L_\phi}.$$

The left-hand side of (15) represents the time-averaged value of a weighted combination of $\mathbb{E}[\Delta_\phi^t]$ and $\mathbb{E}[\Delta_h^t]$. The convergence rate, dictating how rapidly this value diminishes to zero, is tied to the effective noise variances $\Omega_1^2$ and $\Omega_2^2$. These variances result from the SAM gradient perturbation parameter $\rho^2$ and three stochastic variances $\sigma^2, \sigma_\phi^2$, and $\sigma_h^2$.

## 4 EXPERIMENTS

In this section, we empirically assess the robustness of our pFedSAM against the state-of-the-art black-box and white-box backdoor attacks. For the black-box attack, we choose the BadNet attack Gu et al. (2019), which is the most commonly used attack in centralized training. For the white-box attacks, we implement the DBA Xie et al. (2020) and BapFL Ye et al. (2023) attacks. DBA significantly enhances the persistence and stealthiness against FL on diverse data by breaking down the BadNet trigger pattern into distinct local patterns and injecting them in a distributed way. BapFL Ye et al. (2023) is the most recent backdoor attack specifically tailored for pFL with partial model-sharing. We compare our proposed pFedSAM method with seven widely used defense strategies in FL: Krum Blanchard et al. (2017), Multi-Krum Blanchard et al. (2017), Adding Noise (AD) Du et al. (2019); Sun et al. (2019); Wang et al. (2020), Norm Clipping (NC) Shejwalkar et al. (2022); Sun et al. (2019); Wang et al. (2020), Ditto Li et al. (2021a), FedRep Collins et al. (2021), and Simple-Tuning (ST) Qin et al. (2023). Krum and Multi-Krum aim to identify and filter malicious clients by selecting one or multiple model updates for aggregation based on their similarity. NC and AD aim to mitigate backdoor attacks during the training by limiting the norm of model updates or adding Gaussian noise to model updates before aggregation. We set the threshold $c \in \{0.5, 1.0\}$ in NC and noise scales $\sigma \in \{10^{-5}, 5 \times 10^{-4}\}$ in AD according to Zhang et al. (2022). Ditto is a full model-sharing pFL method that has been demonstrated to provide robustness benefits. FedRep is the most commonly used partial model-sharing pFL method, which has been validated in Qin et al. (2023) to offer superior robustness against black-box attacks compared to other pFL methods. ST is a newly proposed defense method in Qin et al. (2023) that re-initializes and retrains the local linear classifier on benign local dataset at each client while freezing the remaining parameters of its model.

## 4.1 EXPERIMENTAL SETTINGS

We run each experiment 5 times with distinct random seeds, and provide the average testing accuracy and training loss in the same last round for fair comparison. All the algorithms were implemented using PyTorch and executed on an Ubuntu server equipped with four NVIDIA RTX A6000 GPUs.

**Datasets and Models.** Following the prior works in robust FL Xie et al. (2020); Zhang et al. (2023); Wang et al. (2020); Xie et al. (2021); Ye et al. (2023); Qin et al. (2023), we choose the following two common datasets: MNIST LeCun et al. (1998) and CIFAR-10 Krizhevsky et al. (2009). The heterogeneity of these datasets across clients is controlled by following the Dirichlet distribution Hsu et al. (2019) with concentration parameter $\beta$ (default $\beta = 0.5$), where smaller $\beta$ indicates larger data heterogeneity across clients. As with Liang et al. (2020), we use a CNN with two convolutional layers followed by three fully-connected layers for CIFAR10, and an MLP with two hidden layers for MNIST, respectively.

**Attack Setup.** We conduct the FL training over 100 and 300 communication rounds for MNIST and CIFAR10, respectively. We consider the stronger and stealthier backdoor attack setting that malicious clients continuously participate in every round. We consider 100 clients in total by default. Following the existing attack setting Xie et al. (2020); Ye et al. (2023); Qin et al. (2023), we randomly sample 10 clients, including 4 malicious clients for DBA and BapFL or 1 malicious client for BadNet. The rest are benign clients. For pFedSAM, we set the number of local epochs to train the personal parameters to be 2, followed by 2 epochs for the shared parameters in each FL round. The hyper-parameter $\rho$ is set as 0.05 by default. All other methods use the same number of local epochs as pFedSAM. The poison ratio controls the fraction of backdoored samples added per training batch. Malicious clients poison 20 out of 64 samples per batch on CIFAR-10 and MNIST. More details are given in Appendix A.

**Evaluation Metrics.** We utilize two evaluation metrics, attack success rate (ASR) and main task accuracy (ACC), to gauge the effectiveness of pFedSAM. ASR is calculated as the proportion of successfully attacked poisoned samples relative to the total number of poisoned samples. ACC denotes the model's accuracy when tested with benign samples. An effective backdoor attack should achieve a high ASR while maintaining a high ACC, demonstrating its ability to manipulate the model's outputs effectively without compromising its performance on the primary task. Furthermore, to ensure unbiased evaluation, we compute ASR only on samples where the true label differs from the target label Xie et al. (2020).

## 4.2 EXPERIMENTAL RESULTS

We compare pFedSAM with the baselines under the BadNet, DBA, and BapFL attacks.

Table 1: Black-box BadNet attack evaluation

| **Defenses** | MNIST | | CIFAR-10 | |
|---|---|---|---|---|
| | ACC | ASR | ACC | ASR |
| FedAvg (no defense) | **96.09** | 97.03 | 70.94 | 31.88 |
| FedRep | 90.44 | 35.86 | 72.85 | 7.15 |
| Ditto | 87.66 | 58.41 | 72.28 | 30.61 |
| NC ($c = 0.5$) | 95.82 | 98.62 | 56.06 | 19.64 |
| NC ($c = 1.0$) | 95.96 | 98.57 | 69.54 | 22.96 |
| AD ($\sigma = 10^{-5}$) | 95.26 | 96.90 | 70.97 | 20.59 |
| AD ($\sigma = 5 \times 10^{-4}$) | 95.12 | 96.43 | 42.15 | 10.86 |
| Krum | 93.58 | 31.96 | 62.88 | 15.10 |
| Multi-Krum | 95.86 | 19.58 | 69.68 | 15.55 |
| ST | 66.58 | 33.92 | **75.21** | 15.57 |
| pFedSAM | 91.42 | **14.51** | 75.06 | **5.76** |

**BadNet Attack.** As shown in Table 1, BadNet attack achieves more than 97% and 31% ASRs on MNIST and CIFAR-10, respectively. For both datasets, partial model-sharing pFL methods, such as FedRep and pFedSAM effectively reduce the ASR while maintaining high ACC under the black-box setting. Fedrep can effectively reduce the ASR below 36% on MNIST and 8% on CIFAR-10.

Table 2: White-box DBA attack evaluation

| Defenses | MNIST | | CIFAR-10 | |
|---|---|---|---|---|
| | ACC | ASR | ACC | ASR |
| FedAvg (no defense) | **94.54** | 100.00 | 66.32 | 74.82 |
| FedRep | 87.47 | 21.69 | 67.73 | 6.19 |
| Ditto | 56.57 | 85.23 | 60.53 | 15.01 |
| NC ($c = 0.5$) | 92.68 | 97.25 | 65.68 | 37.04 |
| NC ($c = 1.0$) | 92.89 | 99.96 | 66.33 | 44.99 |
| AD ($\sigma = 10^{-5}$) | 93.97 | 99.99 | 57.24 | 34.91 |
| AD ($\sigma = 5 \times 10^{-4}$) | 93.07 | 99.88 | 36.88 | 15.24 |
| Krum | 88.62 | 92.50 | 62.01 | 8.33 |
| Multi-Krum | 94.01 | 97.36 | **69.68** | 20.92 |
| ST | 77.95 | 51.76 | 67.88 | 8.77 |
| pFedSAM | 89.10 | **12.13** | 69.61 | **5.06** |

Conversely, the full model-sharing pFL method Ditto does not significantly enhance the robustness against the backdoor attack. The ASR is only reduced within 1% on CIFAR-10 and remains over 58% on MNIST. This matches the prior results observed in Qin et al. (2023).

NC with $c = 1.0$ brings a slight increase in robustness compared with FedAvg. The ASR is reduced only within 1% on MNIST. Although we can set a small threshold to mitigate the influence of malicious clients, such as $c = 0.5$, it will significantly reduce the ACC because we also limit the contribution from benign clients, such as ACC is below 57% on CIFAR-10. AD yields similar effects on accuracy when employing a large noise scale $\sigma = 5 \times 10^{-4}$, leading to ACC dropping below 43%. By detecting and filtering the potential malicious clients, Krum and Multi-Krum improve the robustness against the backdoor attack. However, their ability to identify attackers is limited, and the ASR remains at least 19% on MNIST and 15% on CIFAR-10. Additionally, the ACC is influenced because these methods may filter potential benign users. Although these NC, AD, Krum, and Multi-Krum exhibit varying robustness improvements against the BadNet attack, they all encounter a significant trade-off between robustness and accuracy. ST can achieve similar ASRs with Krum on both datasets. However, its ACC is unstable and cannot completely eliminate the risk of backdoor attacks.

We can also observe that among all the methods, pFedSAM achieves the best robustness with the lowest ASR while still maintaining a high ACC. Specifically, pFedSAM can reduce the ASR to 5.76% on CIFAR-10 and 14.51% on MNIST, respectively, while achieving a comparable ACC to the highest ACC among all the methods. Compared to FedRep, pFedSAM exhibits a dual enhancement, improving ACC and reducing ASR concurrently. This clearly highlights the benefits of sharpness-aware training in enhancing robustness.

**DBA Attack.** Table 2 shows the effectiveness of pFedSAM in comparison to the baselines for the DBA attack on both datasets. The ASR is 3%-42% higher than BadNet attack on FedAvg shows that DBA is a more aggressive attack than BadNet. Partial model-sharing pFL methods still demonstrate significant robustness against DBA attack, while full model-sharing pFL does not notably enhance robustness. Specifically, Fedrep can effectively reduce the ASR below 22% on MNIST and 7% on CIFAR-10. Although Ditto reduces ASR to 15.01% on CIFAR-10, its ASR exceeds 85% on MNIST, and the ACCs are low on both datasets.

NC fails to effectively mitigate DBA attack on both datasets. Even with a small clipping threshold of $c = 0.5$, its ASR remains above 97% on MNIST and 37% on CIFAR-10, respectively. AD faces a trade-off between robustness and ACC. Although larger noise $\sigma = 5 \times 10^{-4}$ can reduce ASR, as seen 15.24% on CIFAR-10, it significantly impacts ACC below 37% as it restricts the contribution from benign clients. Krum and Multi-Krum can all partially mitigate the impact of backdoored models, particularly on the CIFAR-10 dataset. Krum achieves an impressive reduction to as low as 8.33% ASR on CIFAR-10. However, they turn to be ineffective on the MNIST dataset as the DBA is considered very strong on this dataset, validating the claim that these methods could fail under the strong attack scenarios. Note that ST achieves a comparable ASR to Krum on CIFAR-10. However,

Table 3: White-box BapFL attack evaluation

| Defenses | MNIST | | CIFAR-10 | |
|---|---|---|---|---|
| | ACC | ASR | ACC | ASR |
| FedRep | 76.77 | 37.50 | 77.50 | 53.13 |
| pFedSAM | **80.36** | 29.17 | 78.87 | 40.91 |
| pFedSAM + NC ($c = 0.5$) | 78.45 | 16.67 | **81.79** | 21.53 |
| pFedSAM + NC ($c = 1.0$) | 75.54 | 18.93 | 81.70 | 29.12 |
| pFedSAM + AD ($\sigma = 10^{-5}$) | 79.05 | 19.50 | 80.91 | 26.85 |
| pFedSAM + AD ($\sigma = 5 \times 10^{-4}$) | 80.17 | 29.36 | 33.83 | 6.70 |
| pFedSAM + Krum | 57.15 | 5.81 | 76.66 | 3.55 |
| pFedSAM + Multi-Krum | 75.46 | **3.66** | 80.97 | **2.82** |
| pFedSAM + ST | 44.76 | 7.04 | 70.38 | 19.67 |

its ACC is unstable and unable to completely mitigate the risks associated with backdoor attacks on the MNIST dataset.

We observe that pFL methods with partial model-sharing, such as FedRep and pFedSAM, can largely defend against DBA attack on both datasets with a much lower ASR. pFedSAM demonstrates superior robustness by achieving the lowest ASR while maintaining a high ACC. In addition, pFedSAM consistently outperforms FedRep in terms of both ACC and ASR. Specifically, pFedSAM demonstrates a 1%-2% increase in ACC and a 1%-9% reduction in ASR compared to FedRep. This highlights the benefits of sharpness-aware training in enhancing the robustness against backdoor attacks and improving model performance on benign samples.

**BapFL Attack.** As the BapFL attack is custom-designed for partial model-sharing pFL, we evaluate this attack only on two methods for comparison: FedRep and pFedSAM. Moreover, we show the flexibility of pFedSAM in incorporating other defense strategies to achieve even better robustness. The results are shown in Table 3. Due to its ASR being 15%-42% higher than DBA attack on FedRep, and DBA being stronger than BadNET, it clearly shows that BapFL is more aggressive than DBA and BadNet under the pFL setting.

From the table, we can observe that the straightforward implementation of pFL with partial model-sharing, FedRep, remains susceptible to BapFL attack with relatively high ASRs, such as 53% on CIFAR-10 and 37.50%. Compared with FedRep, pFedSAM can enhance robustness and accuracy simultaneously under BapFL due to the use of sharpness-aware training in updating the shared parameters, but it still cannot provide an effective defense by itself. Therefore, we combine pFedSAM with the existing defense strategies (NC, AD, Krum, Multi-Krum, and ST) by applying them to the global aggregation step of the shared parameters in pFedSAM to see the effectiveness. From the table, we can observe that pFedSAM in conjunction with AD, NC, or ST does not effectively mitigate the BapFL attack. However, BapFL can be effectively mitigated when integrating the Krum or Multi-Krum into our proposed pFedSAM. Specifically, by integrating pFedSAM with Multi-Krum, it can reduce the ASR to below 4% on both datasets and keep the ACC degradation within 1%.

## 5 CONCLUSION

In this paper, we have introduced pFedSAM, a novel pFL method that provides robustness against both black-box and white-box backdoor attacks. We have shown that pFedSAM can result personalized models with low accuracy on poisoned samples and high accuracy on benign samples under the strong continuous backdoor attack setting. We have also proved the convergence of pFedSAM under the general non-convex and non-IID data distribution setting. Our proposed method is flexible and can be integrated with existing defense strategies to defend against more aggressive attacks. For future work, we will consider other implementations of the SAM optimizer and conduct more comprehensive experiments on a wide range of datasets, tasks, and attack settings.

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

## A  IMPLEMENTATION DETAILS

### A.1  ATTACK SETTING

**BadNet.** BadNet trigger is the most commonly used attack in centralized training. We employ a continuous attack setup where the attacker engages in attacks throughout the entire training process, from the first round to the last. As a black-box attack, we set the same learning rate and poisoning epoch as benign clients.

**DBA.** DBA uses the same trigger as BadNet but in a distributed way. Specifically, DBA breaks down a central trigger pattern into distinct local patterns and then integrates each of these patterns into the training dataset of various adversarial entities. Compared to traditional centralized backdoors, DBA exhibits significantly enhanced persistence and stealthiness against FL on diverse data. We use the same continuous attack setting as BadNet in our paper. The poison learning rate/epoch is 0.05/4 for CIFAR-10 and 0.05/10 for MNIST.

**BapFL.** BapFL is a SOTA partial model-sharing backdoor attack in FL. The key principle of BapFL is to preserve clean local parameters while embedding the backdoor into the global parameters. Specifically, BapFL divides the local samples in each batch into a clean set and a poison set. During the attack, the local full model is trained using the clean set, while the global shared parameters are updated exclusively using the poison set. We use the same learning rate and batch size as benign clients. As backdoor attackers start attacking when the ACC converges Ye et al. (2023), we utilize pre-trained models for FedRep and pFedSAM and set the learning rate as 0.01 for benign clients and 0.05 for malicious clients.

### A.2  IMPLEMENTATION DETAILS OF BASELINES

Following the setting in Collins et al. (2021), we set the weights and biases of the final fully-connected layer in each model as the local parameters. Other parts are global shared model parameters. For all baselines, we use the same local sample batch size 64.

**FedAvg.** We set the learning rate to 0.1 and the number of epochs to 2 for each client on both datasets.

**Ditto.** We set the hyper-parameter $\lambda = 0.1$ and learning rate as 0.05 and epoch as 2 for each client.

**NC, AD, Krum, Multi-Krum, and ST.** All these mechanisms follow the same setting as FedAvg.

## B  CONVERGENCE ANALYSIS OF PFEDSAM

### B.1  NOTIONS

For ease of notion, let $\tilde{\phi}_i^{t,s}$, $\tilde{h}_i^{t,s}$ denote the virtual sequences as the SAM/SGD updates following Algorithm 1, regardless of whether they are selected. Thus, for the selected client $i \in \mathcal{S}^t$, we have

$h_i^{t,s} = \tilde{h}_i^{t,s}$ and $\phi_i^{t,s} = \tilde{\phi}_i^{t,s}$. Note that the random variables $\tilde{\phi}_i^{t,s}, \tilde{h}_i^{t,s}$ are independent of the random selection $\mathcal{S}^t$. Then, we have the following update roles for selected clients $i \in \mathcal{S}^t$ in Algorithm 1 as follows

$$h_i^{t+1} = h_i^t - \eta_h \sum_{s=0}^{\tau_h - 1} \widetilde{\nabla}_h F_i(\phi_i^t, \tilde{h}_i^{t,s}),$$

$$\phi_i^{t+1} = \phi_i^t - \eta_\phi \sum_{s=0}^{\tau_\phi - 1} \widetilde{\nabla}_\phi F_i(\tilde{\phi}_i^{t,s} + \epsilon(\tilde{\phi}_i^{t,s}), \tilde{h}_i^{t+1}).$$

where $\epsilon(\tilde{\phi}_i^{t,s})$ is given by

$$\epsilon(\tilde{\phi}_i^{t,s}) = \rho \frac{\widetilde{\nabla}_\phi F_i(\tilde{\phi}_i^{t,s}, \tilde{h}_i^{t+1})}{\left\| \widetilde{\nabla}_\phi F_i(\tilde{\phi}_i^{t,s}, \tilde{h}_i^{t+1}) \right\|_2}.$$

The server update rule is given by

$$\phi^{t+1} = \phi^t - \frac{\eta_\phi}{rN} \sum_{i \in \mathcal{S}^t} \widetilde{\nabla}_\phi F_i(\tilde{\phi}_i^{t,s} + \epsilon(\tilde{\phi}_i^{t,s}), \tilde{h}_i^{t+1}).$$

We use the notation $\widetilde{\Delta}_\phi^t$ as the analogue of $\Delta_\phi^t$ with the virtual variable $\widetilde{H}^{t+1}$.

### B.1.1 USEFUL LEMMAS

**Lemma 1** (Jensen's inequality). *For arbitrary set of $n$ vectors $\{a_i\}_i^n$, $a_i \in \mathbb{R}^d$ and positive weights $\{w_i\}_{i \in [n]}$, $\sum_i^n w_i = 1$,*

$$\left\| \sum_{i=1}^n w_i a_i \right\|^2 \leq \sum_{i=1}^n w_i \|a_i\|$$

**Lemma 2** (Cauchy-Schwarz inequality). *For arbitrary set of $n$ vectors $\{a_i\}_{i=1}^n$, $a_i \in \mathbb{R}^d$,*

$$\left\| \sum_{i=1}^n a_i \right\|^2 \leq n \sum_{i=1}^n \|a_i\|^2$$

**Lemma 3.** *For given two vectors $a, b \in \mathbb{R}^d$,*

$$2\langle a, b \rangle \leq \gamma \|a\|^2 + \gamma^{-1} \|b\|^2, \forall \gamma \geq 0.$$

**Lemma 4** (Bounded $\mathcal{T}_{1,\phi}$). *For $\mathcal{T}_{1,\phi}$, we have,*

$$\mathbb{E}_t[\mathcal{T}_{1,\phi}] \leq -\frac{\eta_\phi \tau_\phi}{2} \mathbb{E}_t \left\| \nabla_\phi F(\phi^t, \widetilde{H}^{t+1}) \right\|^2 + \frac{2\eta_\phi L_\phi^2}{N} \sum_{i=1}^N \sum_{s=0}^{\tau_\phi - 1} \left[ \mathbb{E}_t \left\| \tilde{\phi}_i^{t,s} - \phi^t \right\|^2 + \rho^2 \right].$$

*Proof.* For client $i \in \mathcal{S}^t$, we have $\tilde{\phi}_i^{t,s} = \phi_i^{t,s}$. Thus, we have

$$\mathbb{E}_t[\mathcal{T}_{1,\phi}] = -\eta_\phi \left\langle \nabla_\phi F(\phi^t, \tilde{H}^{t+1}), \frac{1}{rN} \sum_{i \in \mathcal{S}^t} \sum_{s=0}^{\tau_\phi - 1} \widetilde{\nabla}_\phi F_i(\tilde{\phi}_i^{t,s} + \epsilon(\tilde{\phi}_i^{t,s}), \tilde{h}_i^{t+1}) \right\rangle$$

Note that $\tilde{\phi}_i^{t,s}$ is independent of $\mathcal{S}^t$. Then, we have

$$
\begin{aligned}
\mathbb{E}_t[\mathcal{T}_{1,\phi}] &= -\eta_\phi \mathbb{E}_t \Big\langle \nabla_\phi F(\phi^t, \tilde{H}^{t+1}), \frac{1}{N} \sum_{i=1}^{N} \sum_{s=0}^{\tau_\phi - 1} \nabla_\phi F_i(\tilde{\phi}_i^{t,s} + \epsilon(\tilde{\phi}_i^{t,s}), \tilde{h}_i^{t+1}) \Big\rangle \\
&= -\eta_\phi \sum_{s=0}^{\tau_\phi - 1} \mathbb{E}_t \Big\langle \nabla_\phi F(\phi^t, \tilde{H}^{t+1}), \big(\frac{1}{N} \sum_{i=1}^{N} \big(\nabla_\phi F_i(\tilde{\phi}_i^{t,s} + \epsilon(\tilde{\phi}_i^{t,s}), \tilde{h}_i^{t+1}) \\
&\qquad - \nabla_\phi F(\phi^t, \tilde{H}^{t+1}) + \nabla_\phi F(\phi^t, \tilde{H}^{t+1})\big)\big) \Big\rangle \\
&= -\eta_\phi \tau_\phi \mathbb{E}_t \big\| \nabla_\phi F(\phi^t, \tilde{H}^{t+1}) \big\|^2 \\
&\qquad - \eta_\phi \sum_{s=0}^{\tau_\phi - 1} \mathbb{E}_t \Big\langle \nabla_\phi F(\phi^t, \tilde{H}^{t+1}), \frac{1}{N} \sum_{i=1}^{N} \big(\nabla_\phi F_i(\tilde{\phi}_i^{t,s} + \epsilon(\tilde{\phi}_i^{t,s}), \tilde{h}_i^{t+1}) - \nabla_\phi F(\phi^t, \tilde{H}^{t+1})\big) \Big\rangle \\
&\overset{(a)}{\leq} -\frac{\eta_\phi \tau_\phi}{2} \mathbb{E}_t \big\| \nabla_\phi F(\phi^t, \widetilde{H}^{t+1}) \big\|^2 + \frac{\eta_\phi L_\phi^2}{N} \sum_{i=1}^{N} \sum_{s=0}^{\tau_\phi - 1} \Big[ \mathbb{E}_t \big\| \tilde{\phi}_i^{t,s} + \epsilon(\tilde{\phi}_i^{t,s}) - \phi^t \big\|^2 \Big] \\
&\overset{(b)}{\leq} -\frac{\eta_\phi \tau_\phi}{2} \mathbb{E}_t \big\| \nabla_\phi F(\phi^t, \widetilde{H}^{t+1}) \big\|^2 + \frac{2\eta_\phi L_\phi^2}{N} \sum_{i=1}^{N} \sum_{s=0}^{\tau_\phi - 1} \Big[ \mathbb{E}_t \big\| \tilde{\phi}_i^{t,s} - \phi^t \big\|^2 + \rho^2 \Big]
\end{aligned}
$$

where (a) holds due to $\langle x, y \rangle \leq \|x\|^2/2 + \|y\|^2/2$, (b) holds due to $\big\| \epsilon(\tilde{\phi}_i^{t,s}) \big\|^2 \leq \rho^2$. $\qquad \square$

**Lemma 5** (Bounded $\mathcal{T}_{2,\phi}$). *For $\mathcal{T}_{2,\phi}$, we have,*

$$
\begin{aligned}
\mathbb{E}_t[\mathcal{T}_{2,\phi}] &\leq 3L_\phi \eta_\phi^2 \tau_\phi^2 \mathbb{E}_t \big\| \nabla_\phi F(\phi^t, \widetilde{H}^{t+1}) \big\|^2 + \frac{L_\phi \eta_\phi^2 \tau_\phi^2}{rN} \big(\sigma_\phi^2 + 6\delta^2(1 - r)\big) \\
&\qquad + \frac{6L_\phi^2 \eta_\phi^2 \tau_\phi}{N} \sum_{i=1}^{N} \sum_{s=0}^{\tau_\phi - 1} \Big[ \mathbb{E}_t \big\| \tilde{\phi}_i^{t,s} - \phi^t \big\|^2 + \rho^2 \Big].
\end{aligned}
$$

*Proof.* Using $\mathbb{E}\|x\|^2 = \|\mathbb{E}[x]\|^2 + \mathbb{E}\|x - \mathbb{E}[x]\|^2$, we have

$$
\begin{aligned}
\mathbb{E}_t[\mathcal{T}_{2,\phi}] &= L_\phi \mathbb{E}_t \big\| \phi^{t+1} - \phi^t \big\|^2 \\
&= L_\phi \eta_\phi^2 \mathbb{E}_t \Big\| \frac{1}{rN} \sum_{i \in \mathcal{S}^t} \sum_{s=0}^{\tau_\phi - 1} \widetilde{\nabla}_\phi F_i(\tilde{\phi}_i^{t,s} + \epsilon(\tilde{\phi}_i^{t,s}), \tilde{h}_i^{t+1}) \Big\|^2 \\
&= L_\phi \eta_\phi^2 \mathbb{E}_t \Big\| \frac{1}{rN} \sum_{i \in \mathcal{S}^t} \sum_{s=0}^{\tau_\phi - 1} \nabla_\phi F_i(\tilde{\phi}_i^{t,s} + \epsilon(\tilde{\phi}_i^{t,s}), \tilde{h}_i^{t+1}) \Big\|^2 \\
&\quad + L_\phi \eta_\phi^2 \mathbb{E}_t \Big\| \frac{1}{rN} \sum_{i \in \mathcal{S}^t} \sum_{s=0}^{\tau_\phi - 1} \Big[ \widetilde{\nabla}_\phi F_i(\tilde{\phi}_i^{t,s} + \epsilon(\tilde{\phi}_i^{t,s}), \tilde{h}_i^{t+1}) - \nabla_\phi F_i(\tilde{\phi}_i^{t,s} + \epsilon(\tilde{\phi}_i^{t,s}), \tilde{h}_i^{t+1}) \Big] \Big\|^2 \\
&\leq L_\phi \tau_\phi \eta_\phi^2 \sum_{s=0}^{\tau_\phi - 1} \mathbb{E}_t \underbrace{\Big\| \frac{1}{rN} \sum_{i \in \mathcal{S}^t} \nabla_\phi F_i(\tilde{\phi}_i^{t,s} + \epsilon(\tilde{\phi}_i^{t,s}), \tilde{h}_i^{t+1}) \Big\|^2}_{\mathcal{T}_s'} + \frac{L_\phi \eta_\phi^2 \tau_\phi^2 \sigma_\phi^2}{rN}
\end{aligned}
$$

For $\mathcal{T}_s'$, we have

$$\mathcal{T}_s' \leq 3 \underbrace{\left\| \frac{1}{rN} \left( \sum_{i \in \mathcal{S}^t} \nabla_\phi F_i(\tilde{\phi}_i^{t,s} + \epsilon(\tilde{\phi}_i^{t,s}), \tilde{h}_i^{t+1}) - \nabla_\phi F_i(\phi^t, \tilde{h}_i^{t+1}) \right) \right\|^2}_{\mathcal{T}_{s,1}'}$$

$$+ 3 \underbrace{\left\| \frac{1}{rN} \sum_{i \in \mathcal{S}^t} \nabla_\phi F_i(\phi^t, \tilde{h}_i^{t+1}) - \nabla_\phi F(\phi^t, \tilde{H}^{t+1}) \right\|^2}_{\mathcal{T}_{s,2}'} + 3 \left\| \nabla_\phi F(\phi^t, \tilde{H}^{t+1}) \right\|^2.$$

For $\mathcal{T}_{s,1}'$, according to Lemma 1 and Assumption 1, then we take an expectation over the sampling of devices to get

$$\mathbb{E}_t[\mathcal{T}_{s,1}'] \leq \frac{2L_\phi}{N} \sum_{i=1}^{N} \left[ \mathbb{E}_t \left\| \tilde{\phi}_i^{t,s} - \phi^t \right\|^2 + \rho^2 \right]$$

For $\mathcal{T}_{s,2}'$, following Lemma 8 and Assumption 3, we get

$$\mathcal{T}_{s,2}' \leq \left( \frac{N - rN}{N - 1} \right) \frac{1}{rN^2} \sum_{i=1}^{N} \left\| \nabla_\phi F_i(\phi^t, \tilde{h}_i^{t+1}) - \nabla_\phi F(\phi^t, \tilde{H}^{t+1}) \right\|^2 \leq \frac{2}{rN}(1 - r)\delta^2.$$

To finalize the proof, we substitute these terms into the definitions of $\mathcal{T}_s'$ and $\mathcal{T}_{2,\phi}$. $\qquad\square$

**Lemma 6** (Bounded $\mathcal{T}_{3,\phi}$). *(Claim 8, Pillutla et al. (2022)) For $\mathcal{T}_{3,\phi}$, we have,*

$$\mathbb{E}_t[\mathcal{T}_{3,\phi}] \leq 8\eta_h^2 \tau_h^2 L_h \chi^2 (1 - r)\Delta_h^t + 4\chi^2 \eta_h^2 \tau_h^2 L_h \sigma_h^2 (1 - r).$$

**Lemma 7** (Bounded $\mathcal{T}_H$). *(Claim 9, Pillutla et al. (2022)) Assume that $\eta_h \tau_h L_h \leq 1/8$, we have*

$$\mathbb{E}_t[\mathcal{T}_H] \leq -\frac{\eta_h \tau_h r \Delta_h^t}{8} + \frac{\eta_h^2 \tau_h^2 L_h \sigma_h^2 r}{2} + 4\eta_h^3 L_h \tau_h^2 (\tau_h - 1)\sigma_h^2 r.$$

**Lemma 8** (Sampling Without Replacement). *(Lemma 21, Pillutla et al. (2022)) Let $a_1, \ldots, a_n \in \mathbb{R}^d$ be given. Let $S$ be a uniformly random sample of size $m$ from this collection, where the sampling is without replacement. Denoting the mean $\bar{a} = \frac{1}{n} \sum_{i=1}^{n} a_i$, we have*

$$\mathbb{E}_S \left\| \frac{1}{m} \sum_{i \in S} a_i - \bar{a} \right\|^2 \leq \left( \frac{n - m}{n - 1} \right) \frac{1}{nm} \sum_{i=1}^{n} \| a_i - \bar{a} \|^2.$$

**Lemma 9** (Bounded local updates). *Under Assumptions 1, 2, 3, we have*

$$\frac{1}{N} \sum_{i=1}^{N} \mathbb{E}_t \left\| \phi_i^{t,s} - \phi^t \right\|^2 \leq 6\tau_\phi \eta_\phi^2 L_\phi^2 \rho^2 + 18\eta_\phi^2 \tau_\phi^2 \left( \sigma_\phi^2 + \delta^2 + \frac{1}{N} \sum_{i=1}^{N} \| \nabla_\phi F(\phi^t, H^{t+1}) \|^2 \right).$$

*Proof.*

$$\frac{1}{N} \sum_{i=1}^{N} \mathbb{E}_t \left\| \phi_i^{t,s} - \phi^t \right\|^2 = \frac{1}{N} \sum_{i=1}^{N} \mathbb{E}_t \left\| \phi_i^{t,s-1} - \eta_\phi \widetilde{\nabla}_\phi F_i(\phi_i^{t,s-1} + \epsilon(\phi_i^{t,s-1}), h_i^{t+1}) - \phi^t \right\|^2$$

$$= \frac{1}{N} \sum_{i=1}^{N} \left\| \phi_i^{t,s-1} - \phi^t - \eta_\phi \Big( \widetilde{\nabla}_\phi F_i(\phi_i^{t,s-1} + \epsilon(\phi_i^{t,s-1}), h_i^{t+1}) - \widetilde{\nabla}_\phi F_i(\phi_i^{t,s-1}, h_i^{t+1}) \right.$$

$$+ \widetilde{\nabla}_\phi F_i(\phi_i^{t,s-1}, h_i^{t+1}) - \nabla_\phi F_i(\phi_i^{t,s-1}, h_i^{t+1}) + \nabla_\phi F_i(\phi_i^{t,s-1}, h_i^{t+1})$$

$$\left. - \nabla_\phi F(\phi^t, H^{t+1}) + \nabla_\phi F(\phi^t, H^{t+1}) \Big) \right\|^2$$

$$\leq \mathcal{T}_{2,\phi}'' + \mathcal{T}_{2,\phi}'''$$

where

$$\mathcal{T}_{2,\phi}'' = (1 + \frac{1}{2\tau_\phi - 1}) \frac{1}{N} \sum_{i=1}^{N} \|\phi_i^{t,s-1} - \phi^t - \eta_\phi \Big( \widetilde{\nabla}_\phi F_i(\phi_i^{t,s-1} + \epsilon(\phi_i^{t,s-1}), h_i^{t+1}) - \widetilde{\nabla}_\phi F_i(\phi_i^{t,s-1}, h_i^{t+1}) \Big) \|^2,$$

and

$$\mathcal{T}_{2,\phi}''' = \frac{2\tau_\phi \eta_\phi^2}{N} \sum_{i=1}^{N} \|\widetilde{\nabla}_\phi F_i(\phi_i^{t,s-1}, h_i^{t+1}) - \nabla_\phi F_i(\phi_i^{t,s-1}, h_i^{t+1}) + \nabla_\phi F_i(\phi_i^{t,s-1}, h_i^{t+1})$$
$$- \nabla_\phi F(\phi^t, H^{t+1}) + \nabla_\phi F(\phi^t, H^{t+1}) \Big) \|^2$$

For $\mathcal{T}_{2,\phi}''$, we have

$$\mathcal{T}_{2,\phi}'' \le (1 + \frac{1}{2\tau_\phi - 1}) \frac{1}{N} \sum_{i=1}^{N} (\mathbb{E} \|\phi_i^{t,s-1} - \phi^t\|^2 + \eta_\phi^2 L_\phi^2 \rho^2).$$

For $\mathcal{T}_{2,\phi}'''$, we have

$$\mathcal{T}_{2,\phi}''' \le 6\tau_\phi \eta_\phi^2 \Big( \sigma_\phi^2 + \delta^2 + \frac{1}{N} \sum_{i=1}^{N} \|\nabla_\phi F(\phi^t, H^{t+1})\|^2 \Big).$$

Thus, the recursion from $s = 0$ to $\tau_\phi - 1$ generate

$$\frac{1}{N} \sum_{i=1}^{N} \mathbb{E}_t \|\phi_i^{t,s} - \phi^t\|^2 \le \sum_{s=0}^{\tau_\phi - 1} (1 + \frac{1}{2\tau_\phi - 1})^s \Big[ (1 + \frac{1}{2\tau_\phi - 1}) \eta_\phi^2 L_\phi^2 \rho^2 + \mathcal{T}_{2,\phi}''' \Big]$$

$$\le (2\tau_\phi - 1) \Big[ (1 + \frac{1}{2\tau_\phi - 1})^{\tau_\phi - 1} \Big] \Big[ (1 + \frac{1}{2\tau_\phi - 1}) \eta_\phi^2 L_\phi^2 \rho^2 + \mathcal{T}_{2,\phi}''' \Big]$$

$$\overset{(a)}{\le} 3\tau_\phi \Big( \mathcal{T}_{2,\phi}''' + 2\eta_\phi^2 L_\phi^2 \rho^2 \Big)$$

$$\le 6\tau_\phi \eta_\phi^2 L_\phi^2 \rho^2 + 18 \eta_\phi^2 \tau_\phi^2 \Big( \sigma_\phi^2 + \delta^2 + \frac{1}{N} \sum_{i=1}^{N} \|\nabla_\phi F(\phi^t, H^{t+1})\|^2 \Big),$$

where (a) holds due to $1 + \frac{1}{2\tau_\phi - 1} \le 2$ and $(1 + \frac{1}{2\tau_\phi - 1})^{\tau_\phi} \le \sqrt{5} < \frac{5}{2}$ for any $\tau_\phi \ge 1$. $\qquad \square$

## B.2 DETAILED PROOF

*Proof.* We start with
$$\mathbb{E}_t[F(\phi^{t+1}, H^{t+1}) - F(\phi^t, H^t)] = \underbrace{\mathbb{E}_t[F(\phi^{t+1}, H^{t+1}) - F(\phi^t, H^{t+1})]}_{\mathcal{T}_\phi}$$
$$+ \underbrace{\mathbb{E}_t[F(\phi^t, H^{t+1}) - F(\phi^t, H^t)]}_{\mathcal{T}_H}$$

For $\mathcal{T}_\phi$, we have

$$\mathbb{E}_t[F(\phi^{t+1}, H^{t+1}) - F(\phi^t, H^{t+1})] \overset{(a)}{\le} \langle \nabla_\phi F(\phi^t, H^{t+1}), \phi^{t+1} - \phi^t \rangle + \frac{L_\phi}{2} \mathbb{E}_t \|\phi^{t+1} - \phi^t\|^2$$

$$= \langle \nabla_\phi F(\phi^t, H^{t+1}) - \nabla_\phi F(\phi^t, \widetilde{H}^{t+1}), \phi^{t+1} - \phi^t \rangle$$
$$+ \langle \nabla_\phi F(\phi^t, \widetilde{H}^{t+1}), \phi^{t+1} - \phi^t \rangle + \frac{L_\phi}{2} \mathbb{E}_t \|\phi^{t+1} - \phi^t\|^2$$

$$\overset{(b)}{\le} \langle \nabla_\phi F(\phi^t, \widetilde{H}^{t+1}), \phi^{t+1} - \phi^t \rangle + L_\phi \mathbb{E}_t \|\phi^{t+1} - \phi^t\|^2$$
$$+ \frac{1}{2L_\phi} \Big\| \nabla_\phi F(\phi^t, H^{t+1}) - \nabla_\phi F(\phi^t, \widetilde{H}^{t+1}) \Big\|^2$$

$$\overset{(c)}{\le} \underbrace{\langle \nabla_\phi F(\phi^t, \widetilde{H}^{t+1}), \phi^{t+1} - \phi^t \rangle}_{\mathcal{T}_{1,\phi}} + \underbrace{L_\phi \mathbb{E}_t \|\phi^{t+1} - \phi^t\|^2}_{\mathcal{T}_{2,\phi}} + \underbrace{\frac{\chi^2 L_h}{2n} \sum_{i=1}^{n} \Big\| \widetilde{h}_i^{t+1} - h_i^{t+1} \Big\|}_{\mathcal{T}_{3,\phi}}$$

where (a) and (c) follow from Assumption 1 and (b) follows from Lemma 3. According to Lemmas 4, 5 and 6, we have

$$
\begin{aligned}
\mathbb{E}_t[F(\phi^{t+1}, H^{t+1}) - F(\phi^t, H^{t+1})] \leq {}& \left(-\frac{\eta_\phi \tau_\phi}{2} + 3L_\phi \eta_\phi^2 \tau_\phi^2\right) \mathbb{E}_t[\widetilde{\Delta}_\phi^t] \\
& + \frac{6L_\phi^2 \eta_\phi^2 \tau_\phi + \eta_\phi L_\phi^2}{N} \sum_{i=1}^N \sum_{s=0}^{\tau_\phi - 1} \left[\mathbb{E}_t\left\|\tilde{\phi}_i^{t,s} - \phi^t\right\|^2 + \rho^2\right] + 4\chi^2 \eta_h^2 \tau_h^2 L_h \sigma_h^2 (1 - r) \\
& + \frac{L_\phi \eta_\phi^2 \tau_\phi^2}{rN} \left(\sigma_\phi^2 + 6\delta^2(1 - r)\right) + 8\eta_h^2 \tau_h^2 L_h \chi^2 (1 - r) \Delta_h^t \\
\overset{(a)}{\leq} {}& -\frac{\eta_\phi \tau_\phi}{4} \mathbb{E}_t[\widetilde{\Delta}_\phi^t] + \underbrace{\frac{2\eta_\phi L_\phi^2}{N} \sum_{i=1}^N \sum_{s=0}^{\tau_\phi - 1} \mathbb{E}_t\left\|\tilde{\phi}_i^{t,s} - \phi^t\right\|^2}_{\mathcal{T}_{2,\phi}'} + 4\chi^2 \eta_h^2 \tau_h^2 L_h \sigma_h^2 (1 - r) \\
& + \frac{L_\phi \eta_\phi^2 \tau_\phi^2}{rN} \left(\sigma_\phi^2 + 6\delta^2(1 - r)\right) + 8\eta_h^2 \tau_h^2 L_h \chi^2 (1 - r) \Delta_h^t + 2\eta_\phi \tau_\phi L_\phi^2 \rho^2,
\end{aligned}
$$

where (a) holds due to $\eta_\phi \leq \min\{1/(12L_\phi \tau_\phi), 1/(6\tau_\phi)\}$ According to Lemma 9, we have

$$
\begin{aligned}
\mathbb{E}_t[F(\phi^{t+1}, H^{t+1}) - F(\phi^t, H^{t+1})] \overset{(a)}{\leq} {}& -\frac{\eta_\phi \tau_\phi}{8} \mathbb{E}_t[\widetilde{\Delta}_\phi^t] + (12\eta_\phi^3 L_\phi^4 \tau_\phi^2 + 2\eta_\phi \tau_\phi L_\phi^2) \rho^2 \\
& + 36\eta_\phi^3 \tau_\phi^3 L_\phi^2 (\sigma_\phi^2 + \delta^2) + 4\chi^2 \eta_h^2 \tau_h^2 L_h \sigma_h^2 (1 - r) \\
& + \frac{L_\phi \eta_\phi^2 \tau_\phi^2}{rN} \left(\sigma_\phi^2 + 6\delta^2(1 - r)\right) + 8\eta_h^2 \tau_h^2 L_h \chi^2 (1 - r) \Delta_h^t
\end{aligned} \tag{16}
$$

where (a) holds due to $16\eta_\phi^2 L_\phi^2 \tau_\phi^2 \leq 1/8$. Combining (16) and Lemma 7, we have

$$
\begin{aligned}
\mathbb{E}_t[F(\phi^{t+1}, H^{t+1}) - F(\phi^t, H^t)] \overset{(a)}{\leq} {}& -\frac{\eta_\phi \tau_\phi}{8} \mathbb{E}_t[\widetilde{\Delta}_\phi^t] - \frac{\eta_h \tau_h r}{16} \mathbb{E}[\Delta_h^t] + (12\eta_\phi^3 L_\phi^4 \tau_\phi^2 + 2\eta_\phi \tau_\phi L_\phi^2) \rho^2 \\
& + 36\eta_\phi^3 \tau_\phi^3 L_\phi^2 (\sigma_\phi^2 + \delta^2) + 4\eta_h^2 \tau_h^2 L_h \sigma_h^2 (r + \chi^2(1 - r)) \\
& + \frac{L_\phi \eta_\phi^2 \tau_\phi^2}{rN} \left(\sigma_\phi^2 + 6\delta^2(1 - r)\right) + \frac{\eta_h^2 \tau_h^2 L_h \sigma_h^2 r}{2}
\end{aligned}
$$

where (a) holds due to $128\eta_\phi L_\phi \tau_\phi \chi^2 (r - 1) \leq 1$. Taking an unconditional expectation, summing it over $t = 0$ to $T - 1$ and rearranging, we get

$$
\begin{aligned}
\frac{1}{T} \sum_{t=1}^{T-1} \left(\frac{\eta_\phi \tau_\phi}{8} \mathbb{E}_t[\widetilde{\Delta}_\phi^t] + \frac{\eta_h \tau_h r}{16} \mathbb{E}[\Delta_h^t]\right) \leq {}& \frac{\Delta F_0}{T} + (12\eta_\phi^3 L_\phi^4 \tau_\phi^2 + 2\eta_\phi \tau_\phi L_\phi^2) \rho^2 + 36\eta_\phi^3 \tau_\phi^3 L_\phi^2 (\sigma_\phi^2 + \delta^2) \\
& + 4\eta_h^2 \tau_h^2 L_h \sigma_h^2 (r + \chi^2(1 - r)) + \frac{L_\phi \eta_\phi^2 \tau_\phi^2}{rN} \left(\sigma_\phi^2 + 6\delta^2(1 - r)\right) + \frac{\eta_h^2 \tau_h^2 L_h \sigma_h^2 r}{2}
\end{aligned} \tag{17}
$$

This is a bound in terms of the virtual iterates $\tilde{H}^{t+1}$. However, we wish to show a bound in terms of the actual iterate $H^t$. Using Lemma 2 and Assumption 1, we have

$$
\begin{aligned}
\mathbb{E}_t[\nabla_\phi F(\phi^t, H^t) - \nabla_\phi F(\phi^t, \tilde{H}^{t+1})] \leq {}& \frac{1}{N} \sum_{i=1}^N \mathbb{E}_t\left\|\nabla_\phi F_i(\phi^t, h_i^t) - \nabla_\phi F_i(\phi^t, \tilde{h}_i^{t+1})\right\|^2 \\
\leq {}& \frac{\chi^2 L_\phi L_h}{N} \sum_{i=1}^N \mathbb{E}_t\left\|\tilde{h}_i^{t+1} - h_i^t\right\|^2 \\
\overset{(a)}{\leq} {}& \frac{\chi^2 L_\phi L_h}{N} \sum_{i=1}^N \left(16\eta_h^2 \tau_h^2 \left\|\nabla_h F_i(\phi^t, h_i^t)\right\|^2 + 8\eta_h^2 \tau_h^2 \sigma_h^2\right) \\
= {}& 8\eta_h^2 \tau_h^2 \sigma_h^2 \chi^2 L_\phi L_h + 16\eta_h^2 \tau_h^2 \chi^2 L_\phi L_h \Delta_h^t,
\end{aligned}
$$

where (a) holds due the Lemma 23 in Pillutla et al. (2022). Using

$$\left\|\nabla_\phi F(\phi^t, H^t)\right\|^2 \leq 2\left\|\nabla_\phi F(\phi^t, H^t) - \nabla_\phi F(\phi^t, \tilde{H}^{t+1})\right\|^2 + 2\left\|\nabla_\phi F(\phi^t, \tilde{H}^{t+1})\right\|^2$$

we have

$$\mathbb{E}[\Delta_\phi^T] \leq 2\mathbb{E}[\widetilde{\Delta}_\phi^t] + 16\eta_h^2\tau_h^2\sigma_h^2\chi^2 L_\phi L_h + 32\eta_h^2\tau_h^2\sigma_h^2\chi^2 L_\phi L_h \mathbb{E}[\Delta_h^t].$$

Thus, when $32\gamma^2\chi^2\alpha \leq \frac{1}{2}$, we have

$$\frac{\eta_\phi\tau_\phi}{16}\mathbb{E}[\Delta_\phi^t] + \frac{\eta_h\tau_h r}{32}\mathbb{E}[\Delta_h^t] \leq \frac{\eta_\phi\tau_\phi}{8}\mathbb{E}[\widetilde{\Delta}_\phi^t] + \frac{\eta_h\tau_h r}{16}\mathbb{E}[\Delta_h^t] + \eta_\phi\tau_\phi\eta_h^2\tau_h^2\sigma_h^2\chi^2 L_\phi L_h.$$

Summing it over $t = 0$ to $T - 1$ and plugging in (17), we get

$$\frac{1}{T}\sum_{t=0}^{T-1}(\frac{\eta_\phi\tau_\phi}{16}\mathbb{E}[\Delta_\phi^t] + \frac{\eta_h\tau_h r}{32}\mathbb{E}[\Delta_h^t])$$

$$\leq \frac{1}{T}\sum_{t=0}^{T-1}(\frac{\eta_\phi\tau_\phi}{8}\mathbb{E}[\widetilde{\Delta}_\phi^t] + \frac{\eta_h\tau_h r}{16}\mathbb{E}[\Delta_h^t]) + \eta_\phi\tau_\phi\eta_h^2\tau_h^2\sigma_h^2\chi^2 L_\phi L_h$$

$$\leq \frac{\Delta F_0}{T} + (12\eta_\phi^3 L_\phi^4\tau_\phi^2 + 2\eta_\phi\tau_\phi L_\phi^2)\rho^2 + 36\eta_\phi^3\tau_\phi^3 L_\phi^2(\sigma_\phi^2 + \delta^2) + \eta_\phi\tau_\phi\eta_h^2\tau_h^2\sigma_h^2\chi^2 L_\phi L_h$$

$$+ 4\eta_h^2\tau_h^2 L_h\sigma_h^2(r + \chi^2(1-r)) + \frac{L_\phi\eta_\phi^2\tau_\phi^2}{rN}\left(\sigma_\phi^2 + 6\delta^2(1-r)\right) + \frac{\eta_h^2\tau_h^2 L_h\sigma_h^2 r}{2}$$

Plugging in $\eta_\phi = \frac{\alpha}{L_\phi\tau_\phi}$ and $\eta_h = \frac{\alpha}{L_h\tau_h}$ completes the proof. $\qquad\square$

