# OpenReview forum: "pFedSAM: Secure Federated Learning Against Backdoor Attacks via Personalized Sharpness-Aware Minimization"
_ICLR.cc/2024/Conference — ICLR 2024 Conference Withdrawn Submission_

### Official Review · Reviewer_ZAXx · 2023-10-19

**Soundness:** 2 fair
**Presentation:** 3 good
**Contribution:** 2 fair
**Rating:** 3
**Confidence:** 4

**Summary:**

This paper introduces pFedSam, a backdoor defense tailored for personalized federated learning, founded on sharpness-aware minimization.

**Strengths:**

* The paper is excellently written and highly accessible.

* The algorithm's convergence property is thoroughly discussed.

* Multiple attacks are taken into consideration.

**Weaknesses:**

* The paper's novelty is somewhat limited, as it combines SAM with FedRep (Liang et al.), which is a straightforward combination. Notably, prior works like (Qu et al., 2022) and (Caldarola et al., 2022) have already demonstrated the successful combination of SAM and federated learning, while (Zhu et al., 2022) has shown SAM's effectiveness in defending against backdoor attacks.

* The experiments lack comprehensiveness, relying solely on two small-scale datasets for evaluation. It is recommended that the authors consider employing a deeper network, such as ResNet18, in the Cifar10 experiments to enable more meaningful accuracy comparisons.

* The experimental results in Tables 1 and 2 reveal that pFedSam only marginally improves ASR compared to FedRep (7.15% to 5.76% in CIFAR10). However, it's worth considering that the significant ASR reduction compared to other personalized solutions, such as Ditto and FedAvg with additional defenses (up to 31% in CIFAR10), may not be attributed to SAM but rather to FedRep. The rationale behind FedRep's effectiveness in defending against backdoor attacks is straightforward: it discards the classifier of the poisoned shared models and trains them individually with benign clients. However, it's essential to investigate whether SAM training over the feature extractor also keeps the representations free from poisoning. This can be easily demonstrated by generating a t-SNE visualization of the representation distribution.

*  Both FedRep and pFedSAM exhibit vulnerability to backdoor attacks in the BapFL attack, necessitating the addition of existing defenses like Krum and NC to restore satisfactory performance. This highlights that integrating SAM alone may not suffice to resist backdoor poisoning, with the majority of ASR reduction likely attributable to other defenses, diminishing pFedSam's value.

* The motivation behind using SAM in personalized federated learning backdoor defense appears somewhat weak and could benefit from further elaboration.

* A critical point to note is the absence of theoretical guarantees regarding the robustness against backdoor attacks.



Qu Z, Li X, Duan R, et al. Generalized federated learning via sharpness aware minimization[C]//International Conference on Machine Learning. PMLR, 2022: 18250-18280.

Caldarola D, Caputo B, Ciccone M. Improving generalization in federated learning by seeking flat minima[C]//European Conference on Computer Vision. Cham: Springer Nature Switzerland, 2022: 654-672.

Zhu M, Wei S, Shen L, et al. Enhancing Fine-Tuning Based Backdoor Defense with Sharpness-Aware Minimization[J]. arXiv preprint arXiv:2304.11823, 2023.

**Questions:**

The paper's results indicate that FedRep performs well in most attack settings. Consequently, the underlying significance of investigating backdoor defense in personalized federated learning appears limited and less engaging.

I am inclined to vote for rejection for this paper unless most of my concerns are addressed, given that the novelty, motivation, comprehensivenss of experiment, and the experiment results are all weak.

---

### Official Review · Reviewer_ByDC · 2023-11-01

**Soundness:** 2 fair
**Presentation:** 2 fair
**Contribution:** 2 fair
**Rating:** 3
**Confidence:** 5

**Summary:**

This paper proposes a new federated learning method, named pFedSAM, to defense against backdoor attacks. It is achieved by two key modules: partial model personalization and sharpness-aware training. The partial model personalization lets each client own its locally preserved linear classifier to block the propagation of backdoor features from malicious clients to benign clients. The sharpness-aware training generates local flat model updates with better stability and perturbation resilience, resulting in a globally flat model that is robust to the injection of backdoor features from malicious clients. Extensive experiments show the superiority of pFedSAM over state-of-the-art robust baselines in terms of both robustness and accuracy.

**Strengths:**

1.	The investigated problem is interesting, which has not been well addressed by existing works in personalized federated learning.
2.     Convergence analysis .

**Weaknesses:**

1. The related work cited in the introduction, such as Krum, Bulyan and FoolsGold, does not represent the most cutting-edge research. Additionally, there have been numerous advanced defense strategies proposed in the existing literature, e.g., [1], [2], [3], [4].

2. It is essential to provide a more detailed description of your motivation for this work. Furthermore, it is important to enhance the articulation of the advantages your method offers in comparison to other existing approaches. Additionally, explaining the reasons behind your method's effectiveness in countering both black-box and white-box backdoor attacks, will contribute to a more comprehensive understanding of your research.

3. The defenses evaluated in this paper are not state-of-the-art. Although the authors have explored seven representative defenses in this paper, they are not state-of-the-art schemes. Defenses in the following references [1-4] still need to be included and verified.

4. The defense is not tested on advanced poisoning attack [5] (backdoor attack is a type of targeted poisoning attack).

5. Figure 1 does not effectively emphasize the central focus of this article's research.

[1] Phillip Rieger, Thien Duc Nguyen, Markus Miettinen, and Ahmad-Reza Sadeghi. Deepsight: Mitigating backdoor attacks in federated learning through deep model inspection. 2022.
[2] Chulin Xie, Minghao Chen, Pin-Yu Chen, and Bo Li. Crfl: Certifiably robust
federated learning against backdoor attacks. In International Conference on Machine
Learning, pages 11372–11382. PMLR, 2021.
[3] Thien Duc Nguyen, Phillip Rieger, Huili Chen, Hossein Yalame, Helen Möllering, Hossein Fereidooni, Samuel Marchal, Markus Miettinen, Azalia Mirhoseini, Shaza Zeitouni, et al. Flame: Taming backdoors in federated learning. In 31st USENIX Security Symposium, 2022.
[4] Zaixi Zhang, Xiaoyu Cao, Jinyuan Jia, and Neil Zhenqiang Gong. Fldetector: Defending federated learning against model poisoning attacks via detecting malicious clients. In Proceedings of the 28th ACM SIGKDD Conference on Knowledge Discovery and Data Mining, pages 2545–2555, 2022.
[5]Virat Shejwalkar and Amir Houmansadr. Manipulating the byzantine: Optimizing model poisoning attacks and defenses for federated learning. In NDSS, 2021

**Questions:**

see weaknesses

---

### Official Review · Reviewer_DHbL · 2023-11-01

**Soundness:** 3 good
**Presentation:** 3 good
**Contribution:** 3 good
**Rating:** 6
**Confidence:** 4

**Summary:**

This paper introduces pFedSAM, a personalized Federated Learning approach, which integrates:

1. Partial Model Personalization: Allows each client to have its own linear classifier, minimizing the spread of malicious features.
2. Sharpness-aware Training: Ensures stable and resistant model updates against malicious alterations.

pFedSAM effectively counters both black-box and white-box backdoor attacks without compromising model performance. The method's superiority is further validated through extensive evaluations against existing defenses.

**Strengths:**

1. pFedSAM offers protection against both black-box and white-box backdoor attacks, which cover a broad spectrum of attack scenarios in Federated Learning.

2. The paper introduces a combination of Partial Model Personalization and Sharpness-aware Training. This combination not only enhances robustness but also maintains the benign performance of models.

3. The paper provides assurances of convergence for the pFedSAM method even under challenging conditions, like non-convex and non-IID data distributions.

4. Compared to existing robust FL methods, pFedSAM retains or even surpasses accuracy on benign model performances, showcasing its efficacy and practical utility.

**Weaknesses:**

1. Personalized federated learning (pFL) methods like pFedSAM might face scalability issues when deployed in large systems with numerous clients, as individualized model personalization for each client can be resource-intensive.

2. With personalized models, there's an inherent risk of overfitting to local data, which can reduce the generalization capability of the overall federated model.

**Questions:**

1. Are there any computational or resource trade-offs when implementing pFedSAM compared to other federated learning defenses? The paper only emphasize the performance, while the efficiency and the communication costs are important.

2. How does pFedSAM ensure that the sharpness-aware training doesn't introduce vulnerabilities or other issues, especially when confronted by adversaries who have intimate knowledge of this technique?

---

### Official Review · Reviewer_qy2r · 2023-11-10

**Soundness:** 2 fair
**Presentation:** 2 fair
**Contribution:** 1 poor
**Rating:** 3
**Confidence:** 4

**Summary:**

The work focuses on the vulnerability of Federated Learning (FL) to backdoor attacks and proposes to leverage sharpness-aware minimization (SAM) and partial model averaging for decreasing the effectiveness of such attacks (pFedSAM). Theoretical analyses prove the convergence of pFedSAM under non-convex and heterogeneous scenarios. Empirical results show the efficacy of this method.

**Strengths:**

- The paper addresses security concerns of FL scenarios, which are of the utmost importance for the community
- The paper is clear in its notations and definitions.
- The proposed method (pFedSAM) is theoretically analyzed and shown to converge under non-convex and heterogeneous scenarios.
- pFedSAM is compared with current state-of-the-art approaches
- Experiments on pFedSAM show good results
- Experiments settings are provided, as well as necessary resources

**Weaknesses:**

- Missing discussion on important related literature. Specifically, the paper does not discuss existing works using SAM as a defense to backdoor attacks in centralized settings, while other works addressing backdoor attacks in centralized scenarios are only briefly introduced. Examples are [1,2]. How does the literature for centralized settings differ from the FL one? Also, the concept of global sharpness is not discussed, and we know global sharpness helps improve global consistency [3], which could be linked with resilience to backdoor attacks. Additionally, the use of SAM in FL is not discussed. Examples are [3,4,5,6,7,8].
- It is not clear why SAM should be the answer for addressing backdoor attacks, as no intuition or analysis is provided. How do flat minima help in such a case? Why is the model obtained with SAM more robust to backdoor attacks? For instance, [2] shows how backdoor models tend to reach sharper minima (Fig. 1). A similar analysis would be useful to contextualize the paper’s claims.
- I believe the contribution is limited. Specifically, other works already showed the efficacy of SAM for backdoor attacks (e.g., [1][2]) outside the federated scenario, while SAM has already been studied in federated settings, where its efficacy in multiple tasks has been proved (e.g., generalization, differential privacy ecc). Additionally, partial model sharing for handling backdoor attacks is already a well-known technique, as also discussed in the manuscript.
- I believe the experimental setting is not general enough. Specifically, very small models (small CNN and a MLP) are deployed on datasets with easy classification tasks (CIFAR10 and MNIST). How does pFedSAM behave when faced with more complex tasks (e.g., CIFAR100)? And how is the backdoor attack effective when using larger (and more realistic) models (e.g., ResNet)? Additionally, the results are evaluated only under one type of heterogeneity with Dirichlet’s parameter $\beta=0.4$, which does not create a very challenging setting when using such datasets. How does pFedSAM behave under more pronounced heterogeneity, i.e. smaller $\beta$? Lastly, the attacks are not studied under different numbers of attackers or frequency.
- Limitations are not addressed.

**Typos and minor comments**:
- Eq 1: I think $F_i(w_i)$ should be $F_i(w)$ as we are minimizing over $w$.
- The L2 norm is missing in the definition of the SAM gradient (between Eq. 5 and 6 in the text).
- Please consider fixing the use of \cite, \citet and \citep, as the citations often appear in the wrong format, making it hard to read the manuscript. I would suggest changing the color as well.


**References**

[1] Zhu, Mingli, et al. "Enhancing Fine-Tuning Based Backdoor Defense with Sharpness-Aware Minimization." arXiv preprint arXiv:2304.11823 (2023).
[2] Karim, Nazmul, et al. "Efficient Backdoor Removal Through Natural Gradient Fine-tuning." arXiv preprint arXiv:2306.17441 (2023).
[3] Sun, Yan, et al. "Dynamic Regularized Sharpness Aware Minimization in Federated Learning: Approaching Global Consistency and Smooth Landscape." ICML (2023).
[4] Qu, Zhe, et al. "Generalized federated learning via sharpness aware minimization." International Conference on Machine Learning. PMLR, 2022.
[5] Caldarola, Debora, Barbara Caputo, and Marco Ciccone. "Improving generalization in federated learning by seeking flat minima." European Conference on Computer Vision. Cham: Springer Nature Switzerland, 2022.
[6] Shi, Yifan, et al. "Make landscape flatter in differentially private federated learning." Proceedings of the IEEE/CVF Conference on Computer Vision and Pattern Recognition. 2023.
[7] Sun, Yan, et al. "Fedspeed: Larger local interval, less communication round, and higher generalization accuracy." ICLR (2023).
[8] Park, Jinseong, et al. "Differentially Private Sharpness-Aware Training." ICML (2023).

**Questions:**

- How does pFedSAM behave when faced with more complex tasks (e.g., CIFAR100)?
- How is the backdoor attack effective when using larger (and more realistic) models (e.g., ResNet)?
- How does pFedSAM behave under more pronounced heterogeneity, i.e. smaller $\beta$?
- In Tables 1 and 2, pFedSAM is shown to lose accuracy on the MNIST dataset, while the same does not happen on CIFAR10. Do the authors have any intuition on this behavior?
- In Tables 1 and 2, pFedSAM often shows lower accuracy w.r.t. state-of-art-approaches while instead showing lower ASR. Even though this is the hoped result, it is not clear to me why that is happening, i.e. why is SAM more robust than other state-of-the-art methods? And what happens when SAM is combined with other sota?
- What’s the sensibility of pFedSAM to SAM’s hyperparameter $\rho$?
- How sensible is pFedSAM to the varying number of attackers and attack frequency across rounds?